# The association between caesarean section delivery and later life obesity in 21-24 year olds in an Urban South African birth cohort

**Eniola Sogunle** *, **Gwinyai Masukume, Gill Nelson**

School of Public Health, Faculty of Health Sciences, University of the Witwatersrand, Johannesburg, South Africa

* eniolasogunle@yahoo.com

## Abstract

### Background

Obesity is an important public health problem and rates have reached epidemic proportions in many countries. Studies have explored the association between infants delivered by caesarean section and their later life risk of obesity, in many countries outside Africa. As a result of the increasing caesarean section and obesity rates in South Africa, we investigated the association in this country.

### Methods

This was a retrospective analysis of data that were collected from a prospective South African birth cohort (Birth to Twenty Plus), established in 1990. A total of 889 young adults aged 21–24 years were included in the analysis. Poisson regression models were fitted to assess the association between mode of delivery and early adulthood obesity.

### Results

Of the 889 young adults, 106 (11.9%) were obese while 72 (8.1%) were delivered by caesarean section; of which 14 (19.4%) were obese. Caesarean section delivery was significantly associated with obesity in young adults after adjusting for potential confounders like young adults' sex and birth weight, mothers' parity, and education (incidence rate ratio 1.64, 95% CI 1.01–2.68, p = 0.045).

### Conclusion

The association of caesarean section with early adulthood obesity should be interpreted with caution because data on certain key confounding factors such as mothers' pre-pregnancy body mass index and gestational diabetes were not available. Further research from Africa, with larger sample sizes and databases with useful linking of maternal and infant data, should be conducted.

**Data Availability Statement:** There are legal restrictions on sharing the de-identified data set. The primary researcher (ES) signed a confidentiality agreement with DPHRU (MRC

Developmental Pathways for Health Research Unit, Department of Paediatrics, School of Clinical Medicine, University of the Witwatersrand, Johannesburg, South Africa), the gatekeepers of the data. Contact details for access to the data is: Prof Shane Norris (Director: DPHRU, +27 (0)10 447 3721 and Shane.Norris@wits.ac.za).

**Funding:** The author(s) received no specific funding for this work.

**Competing interests:** The authors have declared that no competing interests exist.

## Introduction

Obesity is an important public health problem and rates have reached epidemic proportions in many countries—one in every five young people was estimated to be obese in 2012 in high and middle income countries [1]. Globally, 39% and 13% of individuals aged 18 years and older were overweight and obese, respectively, in 2016 [2]. In South Africa, the rates of overweight or obesity in 2016 were higher among women (61% in women and 31% in men) [3]. Obesity has been associated with adverse health outcomes such as type 2 diabetes, cardiovascular disease, cancer, and premature mortality, in both adults and children [4–8]. Diet (high energy foods), physical inactivity, birth weight, genetics, and parity are commonly explored risk factors of obesity [9–11]. Mode of delivery at birth has also been suggested to be associated with obesity in later life.

Due to the rising rates of obesity, any hypothesized risk factor, such as caesarean section (CS) as a mode of child delivery, is worth exploring. Acknowledging that CSs are sometimes performed to prevent birth complications, biased motives have been identified [12,13] and the use of the procedure is increasing globally [14]. As many as one in three births was reported to be by CS in high-income countries such as the United States of America (USA) in 2017 [15]. The rates of CS in South Africa in 2014, reported in the 2016 South African Health Review, were estimated to be 70.8% in the private health sector, and 24.7% in the public health sector (as reported by the District Health Information System (DHIS) which uses routinely collected health information to manage health services) [16]. An underlying mechanism for the proposed association between CS and later life obesity is the reported limited microbial diversity of offspring delivered by CS [17,18]. This is presumed to persist to adulthood [19].

A number of studies have explored the association between CS and later life obesity. Three systematic reviews and meta-analyses reported increased pooled effect size estimates in young adults (YAs) delivered by CS (such individuals were more likely to be obese) [20–22]. Five additional studies reported similar findings [19,23–26]. However, some studies did not find significant associations, leading to inconsistent findings [27,28]. All previous studies have been conducted in countries outside Africa [19,23–35] and, to our knowledge, nothing is known in the African context. The aim of the study reported in this paper was to investigate the association between CS delivery and early adulthood obesity among singletons in an ongoing longitudinal birth cohort (Birth to Twenty Plus–Bt20+) with more than two decades of follow-up in an urban region in South Africa [36].

## Materials and methods

### Study design and setting, exposure and outcome variables and potential confounders

This was a retrospective analysis of data that were collected from a prospective cohort study established in 1990. Birth to Twenty Plus (formerly Birth to Ten) is an ongoing prospective South African birth cohort established in Soweto, Johannesburg, Gauteng in 1990 [36]. The cohort comprises 3 273 singleton children of mothers who were recruited from antenatal clinics and had an expected delivery date from 23 April to 8 June 1990. Consent to participate in the study was provided by the mothers at enrolment; all subsequent data were also collected with signed consent. Participants have been followed up through administered questionnaires, contact with parents or caregivers, telephone calls, and field visits. Further information on the cohort has been published elsewhere [36,37].

Information on mode of delivery was copied by the investigators of the Bt20+ cohort from the official birth notification forms compiled at the local authority. Information on CS and

vaginal deliveries, i.e. normal (NVD) and assisted (AVD—forceps and vacuum) was available for this study. A digital scale and wall mounted stadiometer were used to measure weight and standing height, to the nearest 0.1 kg and 0.1 cm, respectively. These measurements were taken by trained research personnel. Body Mass Index (BMI) was calculated using the formula weight/height$^2$ (kg/m$^2$). We defined obesity (BMI $\geq$30 kg/m$^2$), as per the World Health Organization (WHO) [38].

We reviewed published literature on factors associated with obesity and mode of delivery, and identified variables of interest from similar studies. Young adults' characteristics included gestational age (at delivery), sex, ethnicity, age, education, alcohol consumption, cigarette smoking, birth weight (low ($<$2.5 kg), normal (2.5–4.0 kg), macrosomic ($>$4 kg)), and breast-feeding duration. Mothers' characteristics included parity, age at delivery, and education. We generated infants' birth weight corrected for gestational age and sex (in centiles) by using the INTERGROWTH-21st calculator to compare these variables with an international standard [39]. Participant's age was calculated as the difference between date of birth and date of data collection.

## Statistical analysis

We assessed the differences in participants' characteristics (early life, young adult and maternal) across the modes of delivery and BMI categories, and compared the sex-stratified prevalence of obesity for each mode of delivery. With a non-rare outcome (obesity) with prevalence of 11.8% for our overall sample, we evaluated the association between CS and early adulthood obesity using Poisson regression models, with robust standard errors. Incidence rate ratio (IRR) and 95% CIs were computed. Although, lifestyle and behavioural characteristics, such as YAs' diet, physical activity, smoking habits, and alcohol consumption, have been associated with obesity, it has been suggested that they are not true confounders in the analysis of the association between mode of delivery and later life obesity [19,40]. Young adults' sex and birth weight in kg, mothers' parity, and education at YA's birth were included in the adjusted models. Although, ethnicity was associated with both the mode of delivery and BMI, it was not included in the analysis. This is because including it did not significantly change the effect estimate, in our final regression model.

Furthermore, we conducted a post-estimation (Wald) test to investigate the differences between obesity rates across sex categories (heterogeneity). This was done by introducing cross product (interaction) terms between mode of delivery and YAs' sex in the final adjusted model. Sex stratified models were then computed. In addition, we analysed the combined VD data to interrogate the potential effect of differential exposure of infants to maternal vaginal and faecal microflora, implicated in the subsequent genesis of childhood obesity, between those born by CS and VD.

## Missing data

In the primary cohort, both BMI (outcome variable) and mode of delivery (primary predictor variable) had missing data of 51.6% and 49.1%, respectively. This was mainly because of loss to follow up and we excluded these individuals from our study. We however, estimated the difference between the primary cohort and our study participants by comparing socio-demographic characteristics between the two groups. Kruskal-Wallis test, Pearson's Chi-square test, and Fisher's exact test were used, where appropriate. Some of the covariates in our final analytic sample had missing values. These variables were young adults' alcohol intake (7.3%), education (1.1%), and birth weight (0.1%); mother's post-school education (3.5%). We assumed that covariates with incomplete data were missing at random, so the probability of a participant

having missing data in these covariates depends on other observed data in the analysis and not on the values of the missing data itself or on unmeasured variables. Multiple imputation, using multivariate normal imputation (MVNI) was performed in Stata to impute missing data for these covariates in the final regression model. The assumption that all variables in the imputation model jointly follow a multivariate normal distribution was not plausible, due to the presence of categorical variables in the model. However, it has been suggested that reasonable inference can still be drawn, even if the assumption of multivariate normality is not held, as this has been demonstrated in different studies [41–43].

The imputation models included variables highlighted for Poisson regression model adjustment, under the statistical analysis section. Twenty imputations were performed and results of the analyses were pooled using Rubin's rules [44]. We compared the observed and imputed data to observe any difference in the mean in both groups. Apart from birth weight (kg), which was a normally distributed continuous variable, all other covariates were in count or categorized form.

## Sensitivity analysis

To investigate the robustness of the inference from the imputed regression analysis and also to address the possibility of residual confounding, sensitivity analysis was conducted. Complete case analysis was performed for regression models–individuals with missing data before imputation were excluded. Models adjusting exclusively for early life factors were computed, as well as those adjusting for lifestyle and behavioural characteristics. We also adjusted for YAs' birth weight as a continuous covariate, and YA's breastfeeding duration at infancy, mother's parity and mother's age at delivery as categorical covariates, in subsequent analyses. P values <0.05 were considered statistically significant. All analyses were conducted using Stata® IC 14 (StataCorp LP College Station, TX).

## Ethical considerations

Prior to analysing the data, we obtained a written ethical approval from the Human Research Ethics Committee of the University of the Witwatersrand (clearance certificate number M161184). Anonymised data were received from the gatekeeper of the primary study and a data sharing agreement was signed to ensure confidentiality and to limit access to the data.

## Results

The study participants comprised 889 21–24 year old YAs after excluding those in the primary cohort with missing data; one cohort member was excluded for being older than 24 (Fig 1). Of the final analytic sample, 72 (8.1%) were delivered by CS of which 14 (19.4%) were obese. The means of the observed and imputed data were similar, hence multiple imputation provided appropriate data for analysis.

To examine the differences in the primary dataset and study participants, Tables 1 and 2 compare the variables in both cohorts. There were significant differences in the distribution of certain characteristics between the primary cohort and study participants, with regards to the YAs' ethnicity, breastfeeding duration at infancy, and smoking, as well as mothers' education, gestational age, and age at YAs birth (P<0.05). With no post school education in both groups, although the rate was higher among the study participants.

Some of the individual distributions across the categories of these covariates followed similar trends between the primary cohort and study participants, although p values show differences statistically (P<0.05). This is because high proportions of Blacks (YA) and non-smokers (YA) were observed in both groups, but they were higher among study participants. High

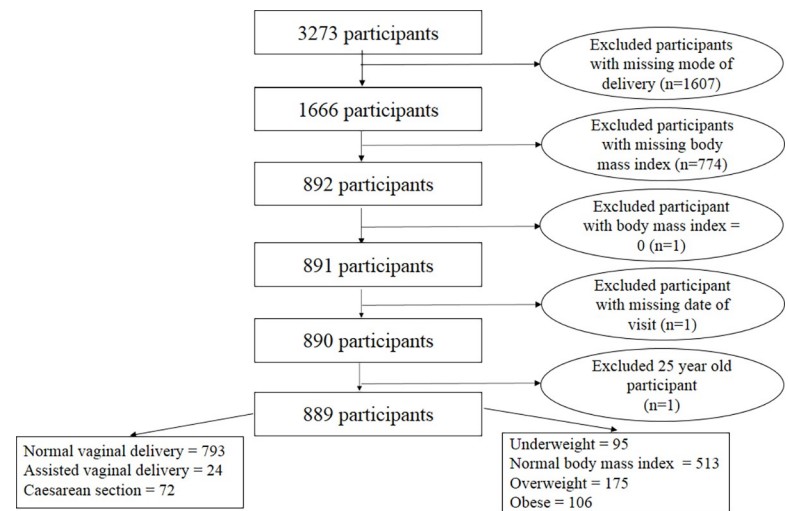

**Fig 1. Flowchart showing the study population and selection of study participants.**

proportions of mothers with no post school education were observed in both groups, and this was also higher among the study participants.

The study participants' characteristics by mode of delivery category are presented in Tables 3 and 4. As seen in Table 3, CSs were more frequent than NVDs in those who had low birth weight or macrosomia, and in Indians and Coloureds (non-Black participants).

No statistically significant differences were observed for gestational age, parity, breastfeeding duration or age at delivery (Table 4).

Compared to the primary cohort, the study participants were more likely to be Black (78.5% vs 90.3%), smokers (19.3% vs 40.4%), have mothers who had no post school education (79.6% vs 92.2%). They were also breast fed for a longer period, and had mothers who had longer gestational age and were younger when they gave birth, than those in the primary cohort. There were no differences between YAs sex, mode of delivery, education, alcohol intake, BMI and birth weight, and mothers' parity, between the primary cohort and the study participants (p>0.05). Certain characteristics were associated with BMI, such as YA's sex (p = <0.001), ethnicity (p = 0.039), smoking habit (p = <0.001), and education (p = <0.001); and mothers' parity at delivery (p = 0.019) (supplementary–S1 and S2 Tables).

Sex-stratified prevalence of obesity are presented in Table 5. Overall, the prevalence of obesity was higher in CS-delivered participants than in those delivered through either AVD or NVD, but there were no differences when the results were stratified by sex.

As seen in Table 6, birth by CS was associated with an almost two-fold increase in the risk of obesity among YAs aged 21–24 years, in the crude analysis. After adjusting for YAs' sex and birth weight, and mothers' parity and education at delivery as potential confounders, a similar estimate was observed.

Although no interaction was found between mode of delivery and either in the final adjusted models, obesity rates were statistically different across the sex (p = <0.001) categories, indicating heterogeneity (see supplementary section, S1 Stata code).

The results of the complete case analysis for regression models are presented in S3 Table in the supplementary information–conclusions were unchanged from these analyses. To address the possibility of residual confounding and to determine if the results held up under different scenarios, a series of sensitivity analyses were conducted and results are presented in S4 Table in the supplementary information. The risk of being obese if delivered by CS were 1.62 times

**Table 1. Comparison between primary cohort and study participants (categorical variables).**

| Variables | Primary cohort | | Study participants | | P value |
|---|---|---|---|---|---|
| | N = 3273 | | N = 889 | | |
| | n | % | n | % | |
| **Young adult characteristics** | | | | | |
| Sex | | | | | 0.511 |
| Male | 1594 | 48.7 | 444 | 49.9 | |
| Female | 1679 | 51.3 | 445 | 50.1 | |
| Ethnicity | | | | | 0.000 |
| Black | 2568 | 78.5 | 803 | 90.3 | |
| Others | 705 | 21.5 | 87 | 9.7 | |
| Mode of delivery | | | | | 0.473 |
| NVD | 1470 | 44.9 | 793 | 89.2 | |
| AVD | 39 | 1.2 | 24 | 2.7 | |
| CS | 157 | 4.8 | 72 | 8.1 | |
| Unknown | 1607 | 49.9 | - | | |
| Smoking | | | | | 0.000 |
| No | 2640 | 80.7 | 530 | 59.6 | |
| Yes | 633 | 19.3 | 359 | 40.4 | |
| Alcohol intake | | | | | 0.845 |
| No | 1004 | 30.7 | 560 | 63.0 | |
| Yes | 482 | 14.7 | 264 | 29.7 | |
| Unknown | 1787 | 54.6 | 65 | 7.3 | |
| Young adult education | | | | | 0.895 |
| <grade12 | 625 | 16.8 | 349 | 39.3 | |
| ≥ grade 12 | 960 | 29.3 | 531 | 59.7 | |
| Unknown | 1688 | 51.6 | 10 | 1.1 | |
| **Early life and maternal characteristics** | | | | | |
| *Mother's post-school education* | | | | | 0.004 |
| No | 2604 | 79.6 | 792 | 92.2 | |
| Yes | 328 | 10.0 | 67 | 7.8 | |
| Unknown | 341 | 10.4 | 30 | 3.4 | |

NVD–normal vaginal delivery, AVD—assisted vaginal delivery, CS—caesarean section

P values approaching or <0.05 shows a difference between the cohorts; Pearson's Chi-square

Missing observations were excluded from the inferential statistics i.e. P value estimation

the risk of being obese if delivered by AVD or NVD. The additional adjustment for YA's breastfeeding duration, smoking habit and alcohol intake, as well as treating certain covariates as linear or categorical in the sensitivity analyses, did not change the conclusions reached from the results of the original analyses. The observed association was not statistically significant when only early life factors (YAs' sex, birth weight and breastfeeding duration; and mothers' parity, gestational age, age and education at delivery) were adjusted for.

## Discussion

### Findings and evidence from previous studies

In this analysis of 21–24 year old South Africans from a longitudinal urban birth cohort, CS was associated with obesity in early adulthood. Among males, CS was statistically associated

**Table 2. Comparison between primary cohort and study participants (continuous variables).**

| Variables | Primary cohort | | Study participants | | P value |
|---|---|---|---|---|---|
| | N = 3273 | | N = 889 | | |
| | Med | IQR | Med | IQR | |
| Young adults age at visit (years) | 23 | 22–23 | 23 | 22–23 | 0.402 |
| Young adults BMI (kg/m$^2$) | 22.1 | 19.7–26.1 | 22.2 | 19.9–26.3 | 0.570 |
| Unknown n(%) | 1728(52.8) | | - | | |
| Mothers gestational age (weeks) | 38.0 | 38.0–40.0 | 38.0 | 38.0–39.0 | 0.000 |
| Unknown n(%) | 102(3.1) | | - | | |
| Young adults birth weight | 3.1 | 2.8–3.4 | 3.1 | 2.8–3.4 | 0.711 |
| Unknown n(%) | 6(0.2) | | 1(0.1) | | |
| Mother's parity at YA birth | 2.0 | 1.0–3.0 | 2.0 | 1.0–3.0 | 0.147 |
| Young adult breastfeeding duration at infancy (months) | 3.5 | 0.0–17.0 | 8.0 | 1.0–20.0 | 0.000 |
| Mother's age at YA delivery (years) | 25.0 | 21.0–30.0 | 24.0 | 20.0–30.0 | 0.002 |
| Unknown n(%) | 2(0.1) | | - | | |

YA—young adult, Med—median, IQR—interquartile range

P values approaching or <0.05 shows a difference between the cohorts; Wilcoxon rank-sum test

Missing observations were excluded from the inferential statistics i.e. P value estimation

with an increased risk of obesity. The magnitude of the association among females is of particular clinical concern. The confidence intervals of the IRRs in the stratified analyses were wide, indicating low precision, due to the small sample size. Most of the sensitivity analyses supported the finding of an association between CS and early adulthood obesity. A marginal statistical significance was observed when early life and maternal characteristics were adjusted for, regardless of their collinearity or correlation with other variables and their effect on the primary predictor.

Significant associations between CS and obesity in early adulthood have been previously reported, despite the inconsistencies in some studies [27,28]. In a systematic review and meta-analysis by Darmasseelane *et al.* (2014), comprising studies published prior to 31 March 2012, the odds of being obese among CS-delivered YAs (aged 18 years or older) from 11 studies, was 22% higher than the odds of being obese among YAs delivered vaginally (OR 1.22, 95% CI 1.05–1.42) [20]. Results from a meta-analysis by Li *et al.* (2013) support this association among YAs aged 19 or older (OR 1.50, 95% CI 1.02–2.20) [21]. The studies included in these reviews were conducted independently in different countries (Finland, England, Scotland, Sweden, Brazil, Netherlands, China, India, and Denmark). Results from a study by Mesquita *et al.* (2013) showed an increase in fat accumulation due to delivery by CS: measured by waist-hip ratio (IRR 1.45, 95% CI 1.18–1.79 [23]. More recently, Yuan *et al.* (2016) and Barros *et al.* (2017) reported increased risks of obesity in CS-delivered YAs (RR 1.15, 95% CI 1.06–1.26, n = 8 486; β coefficient 0.15, 95% CI 0.08–0.23, n = 1 794, respectively) in Boston (USA) and Pelotas (Brazil) [19,26]. Additionally, Hansen *et al.* 2018 reported over a two-fold increase in odds of overweight or obesity in CS–delivered at age 20 years (OR 2.17, 95% CI 1.10, 4.27), after adjustment for potential confounders. This is similar to the risk observed in our study [25].

The aforementioned studies were all conducted outside Africa. However, some of the study sites were in upper-middle income countries, such as Brazil and China which are similar, socio-economically, to South Africa, and hence their results are comparable to those from our study. Also, findings from South Africa, regarding certain established potential confounders, such as socioeconomic status (SES), are different from those reported in high income countries. For example, while most high income countries have reported that high SES leads to

**Table 3. Socio-demographic characteristics by mode of delivery.**

| | Mode of delivery | | | | | | | | P value |
|---|---|---|---|---|---|---|---|---|---|
| | Total | | NVD | | AVD | | CS | | |
| | N = 889 | | n = 793 | | n = 24 | | n = 72 | | |
| | n | % | n | % | n | % | n | % | |
| **Young adult characteristic** | | | | | | | | | |
| *Sex* | | | | | | | | | 0.890 |
| Male | 444 | 49.9 | 398 | 50.2 | 11 | 45.8 | 35 | 48.6 | |
| Female | 445 | 50.1 | 395 | 49.8 | 13 | 54.2 | 37 | 51.4 | |
| *Ethnicity* | | | | | | | | | <0.001 |
| Black | 803 | 90.3 | 727 | 91.7 | 17 | 70.8 | 59 | 81.9 | |
| Non-Black | 86 | 9.7 | 66 | 8.3 | 7 | 29.2 | 13 | 18.1 | |
| *Alcohol intake* | | | | | | | | | 0.617 |
| No | 560 | 63.0 | 502 | 63.3 | 16 | 66.7 | 42 | 58.3 | |
| Yes | 264 | 29.7 | 231 | 29.1 | 8 | 33.3 | 25 | 34.7 | |
| Unknown | 65 | 7.3 | 60 | 7.6 | 0 | - | 5 | 6.9 | |
| *Smoking* | | | | | | | | | 0.479 |
| No | 530 | 59.6 | 472 | 59.5 | 12 | 50.0 | 47 | 59.5 | |
| Yes | 359 | 40.4 | 321 | 40.5 | 12 | 50.0 | 26 | 40.5 | |
| *Education* | | | | | | | | | 0.304 |
| <grade 12 | 349 | 39.3 | 305 | 38.5 | 12 | 50.0 | 32 | 44.4 | |
| ≥ grade 12 | 530 | 59.6 | 479 | 60.4 | 11 | 45.8 | 40 | 55.6 | |
| Unknown | 10 | 1.1 | 9 | 1.1 | 1 | 4.2 | 0 | - | |
| **Early life and maternal Characteristic** | | | | | | | | | |
| *Mother's post-school education* | | | | | | | | | 0.187 [f] |
| No | 792 | 92.2 | 709 | 89.4 | 23 | 95.8 | 59 | 81.9 | |
| Yes | 67 | 7.8 | 59 | 7.4 | 0 | 0.0 | 8 | 11.1 | |
| Unknown | 30 | 3.4 | 25 | 3.2 | 1 | 4.2 | 5 | 6.9 | |
| *Birth weight (kg)* | | | | | | | | | 0.018 [f] |
| LBW (<2.5) | 74 | 8.3 | 63 | 7.9 | 4 | 16.7 | 7 | 9.7 | |
| Normal (2.5–4.0) | 798 | 89.8 | 718 | 90.5 | 20 | 83.3 | 60 | 83.3 | |
| Macrosomia (>4) | 16 | 1.8 | 11 | 1.4 | 0 | - | 5 | 6.9 | |
| Unknown | 1 | 0.1 | 1 | 0.1 | 0 | - | 0 | - | |
| *Birth weight (centile)* | | | | | | | | | |
| SGA (<10th) | 79 | 8.9 | 68 | 8.6 | 5 | 20.8 | 6 | 8.3 | 0.139 [f] |
| AGA (≥10th-<90th) | 680 | 76.5 | 610 | 76.9 | 18 | 75.0 | 53 | 72.2 | |
| LGA (>90th) | 129 | 14.5 | 114 | 14.4 | 1 | 4.2 | 14 | 19.4 | |
| Unknown | 1 | 0.1 | 1 | 0.1 | 0 | - | 0 | - | |

Non-black–Indians and Coloured, LBW–low birth weight, SGA–small for gestational age, AGA–appropriate for gestational age, LGA–large for gestational age, NVD/AVD–Normal/Assisted vaginal delivery

Fisher's exact (f)

higher body mass index, those conducted in South Africa have found that low SES was associated with higher body mass index [45,46].

## Underlying mechanism

It has been suggested that infants born via VD are exposed mainly to microorganisms in the birth canal or vaginal environment, and that those delivered by CS are exposed to micro flora

**Table 4. Maternal characteristics of the study participants by mode of delivery.**

| | Total N = 889 | | NVD n = 793 | | AVD n = 24 | | CS n = 72 | | P value |
|---|---|---|---|---|---|---|---|---|---|
| | Med | IQR | Med | IQR | Med | IQR | Med | IQR | |
| Gestational age (weeks) at delivery | 38.0 | 38.0–39.0 | 38.0 | 38.0–39.0 | 38.0 | 38.0–40.0 | 38.0 | 37.5–39.0 | 0.074 |
| Parity at delivery | 2.0 | 1.0–3.0 | 2.0 | 1.0–3.0 | 1.0 | 1.0–3.0 | 2.0 | 1.0–3.0 | 0.136 |
| Breastfeeding duration (months) | 8.0 | 1.0–20.0 | 9.0 | 1.0–20.0 | 6.5 | 2.5–24.0 | 6.0 | 0.6–18.0 | 0.489 |
| Age at delivery (years) | 24.0 | 20.0–30.0 | 24.0 | 20.0–30.0 | 23.5 | 19.0–30.0 | 25.5 | 21.0–30.0 | 0.541 |

Med–median, IQR–interquartile range, NVD/AVD–Normal/Assisted vaginal delivery, CS–caesarean section

on their mother's skin [47–49]. The abundance of *Coprococcus* and *Ruminococcus* of the family Lachnospiraceae was also reported by Tun *et al.* (2018) in infants delivered through CS–signifying dysbiosis in early life [48]. The investigators further explained the impact of family Lachnospiraceae in promoting adiposity.

Differences in intestinal bacterial colonisation; such as certain *Bifidobacteria spp.* that contribute to digestion and infant intestine development have been reported to be absent in infants delivered by CS [50]. Kalliomäki *et al.* (2008) reported that, compared to those of normal BMI, overweight seven-year old participants had lower Bifidobacteria counts at age one and six years [51]. In addition, infants delivered by CS had almost no *Bifidobacteria spp.* in their faecal samples in studies by Huurre *et al.* (2008) and Biasucci *et al.* (2008) [52,53]. Nonetheless, it is possible that the underlying reason or indication for CS delivery could be a cause of later life obesity. This is because certain clinical states resulting in CS, as well as antibiotics administered during CS, have been suggested to increase the tendency of offspring obesity [40,54,55].

## Limitations and strengths

A limitation of this study is that the exposure and outcome assessments were dependent on the accuracy of the data collected from the Bt20+ cohort. In addition, the current surge of CS

**Table 5. Prevalence of obesity in each mode of delivery category among study participants, by sex.**

| | N | n | % | 95% CI | P value |
|---|---|---|---|---|---|
| *Overall* | | | | | 0.042 |
| NVD | 793 | 88 | 11.1 | 9.1–13.5 | |
| AVD | 24 | 4 | 16.7 | 6.4–37.0 | |
| CS | 72 | 14 | 19.4 | 11.9–30.2 | |
| Total | 889 | 106 | 11.9 | 9.9–14.2 | |
| *Male* | | | | | 0.190 |
| NVD | 398 | 9 | 2.3 | 1.2–4.3 | |
| AVD | 11 | 1 | 9.1 | 1.3–44.0 | |
| CS | 35 | 3 | 8.6 | 2.8–23.5 | |
| Total | 444 | 13 | 2.9 | 1.7–5.0 | |
| *Female* | | | | | 0.133 |
| NVD | 395 | 79 | 20.0 | 16.3–24.2 | |
| AVD | 13 | 3 | 23.1 | 7.6–52.2 | |
| CS | 37 | 11 | 29.7 | 16.8–46.2 | |
| Total | 445 | 93 | 20.9 | 17.4–24.9 | |

CI–confidence interval, VD–vaginal delivery, CS–caesarean section

**Table 6. The association between mode of delivery and early adulthood obesity, stratified by sex and birth weight (using normal BMI as reference)-Imputed models.**

| Variable | IRR | 95%CI | P value | AdjIRR | 95%CI | P value |
|---|---|---|---|---|---|---|
| *Main analysis* | | | | | | |
| crude | | | | | | |
| NVD | 1.00 | reference | | 1.00 | reference | |
| AVD | 1.50 | 0.60–3.76 | 0.384 | 1.41 | 0.57–3.49 | 0.460 |
| CS | 1.75 | 1.05–2.92 | 0.031 | 1.64 | 1.01–2.68 | 0.045 |
| *Male* | | | | | | |
| NVD | 1.00 | | | 1.00 | reference | |
| AVD | 4.02 | 0.56–29.10 | 0.168 | 4.90 | 0.65–37.17 | 0.124 |
| CS | 3.79 | 1.07–13.38 | 0.038 | 4.01 | 1.14–14.09 | 0.031 |
| *Female* | | | | | | |
| NVD | 1.00 | | | 1.00 | reference | |
| AVD | 1.15 | 0.42–3.18 | 0.782 | 1.12 | 0.41–3.07 | 0.717 |
| CS | 1.49 | 0.87–2.54 | 0.146 | 1.44 | 0.85–2.44 | 0.173 |

N = 889; Poisson regression

OR–odds ratio, CI–confidence interval, NVD/AVD–Normal/Assisted vaginal delivery, CS–caesarean section. Adjusted for YAs' sex and birth weight; mothers' parity and education at YA's birth in all models.

in South Africa was not reflected in the results; only 72 (8.1%) women gave birth through CS. Consequently, the 95% CIs were wide, indicating low precision of the ORs. We did not have the relevant data to examine the proposed mechanism underlying the association between CS birth and obesity in later life (i.e. deprivation of CS-born infants of microorganisms essential for regulating digestion). Caesarean delivery was not recorded as being elective or an emergency. It has been proposed that differences in intestine microbiota composition might arise due to prolonged delivery or ruptured fetal membranes [30,56], but this could not be investigated. In addition, the proposed effect (obesity) of antibiotics administered during CS on offsprings delivered through the procedure, could not be assessed due to lack of data.

Management practices in labour and delivery units differ across hospitals and has been reported by Plough *et al.* (2017) to be associated with risk of primary cesarean delivery [57]. The differences in CS rate between public and private hospital could not be explored, as all participants were recruited from public hospital. Finally, key pre-pregnancy information was not available, e.g. mothers' BMI (height and weight); gestational diabetes, preeclampsia, or pregnancy-induced hypertension; smoking habits; information about previous CS; and family income. The absence of these variables might have resulted in residual confounding, leading to higher point estimates than observed in previous studies.

The strength of this analysis was the availability of data to estimate associations between CS and obesity in later life. We were also able to demonstrate temporality as the exposure (CS) preceded the outcome (obesity in early adulthood). Additional strengths of the study were the availability of information on important early life factors of YAs, and prospectively collected data in the cohort. Being able to provide different estimates for normal versus assisted VD is a particular strength of this study. Infant umbilical vein cortisol and other stress molecule concentrations differ significantly between those born by normal and assisted VD, which might influence longer term offspring energy metabolism [58]. Also, higher birth weight, which has been reported to be associated with CS birth, is a risk factor for instrumental VD [59]. Finally, the outcome measure (BMI) and primary predictor (mode of delivery) does not differ between our study participants and primary cohort. Hence, any major effect of selection bias was less likely.

## Conclusions

Caesarean section as a mode of delivery was statistically associated with obesity in the study participants. Further research is required in South Africa, and Africa in general, using routinely collected data that provide useful linking of maternity data with information in other databases. This will help to identify larger study populations and minimize costs, while investigating the association across BMI categories and exploring the underlying mechanisms for the association. Sex-stratified analyses, taking into account the potential interaction with mode of delivery, and differences in obesity rates, should be performed between populations. The reported increased odds of obesity in later life after CS, including those found in this study, support the plausibility of a biological link and should be considered as a motivating factor to reduce CS as a mode of delivery, unless clinically indicated.

## Supporting information

**S1 Stata code. showing heterogeneity in obesity rates between male and female young adults.**
(PDF)

**S2 Stata code. Comparing the mean of observed and imputed data.**
(PDF)

**S1 Table. Body mass index categories of study participants by socio-demographic characteristics.**
(PDF)

**S2 Table. Body mass index categories of study participants by maternal characteristics (continuous).**
(PDF)

**S3 Table. The association between mode of delivery and early adulthood obesity, stratified by sex (using normal BMI as reference)–complete case analysis.**
(PDF)

**S4 Table. Sensitivity analysis: Examining the association between mode of delivery and early adulthood obesity under different scenarios.**
(PDF)

## Acknowledgments

We thank the Developmental Pathways for Health Research Unit (DPHRU) of the University of the Witwatersrand and the investigators of the Birth to Twenty Plus cohort for granting permission to use their data to conduct this study, as well as the study participants without whom this study would not have been possible.

## Author Contributions

**Conceptualization:** Eniola Sogunle, Gwinyai Masukume, Gill Nelson.

**Data curation:** Eniola Sogunle, Gwinyai Masukume, Gill Nelson.

**Formal analysis:** Eniola Sogunle, Gwinyai Masukume.

**Investigation:** Eniola Sogunle, Gwinyai Masukume, Gill Nelson.

**Methodology:** Eniola Sogunle, Gwinyai Masukume, Gill Nelson.

**Project administration:** Eniola Sogunle, Gwinyai Masukume, Gill Nelson.

**Resources:** Eniola Sogunle, Gwinyai Masukume, Gill Nelson.

**Supervision:** Gwinyai Masukume, Gill Nelson.

**Validation:** Eniola Sogunle, Gwinyai Masukume, Gill Nelson.

**Writing – original draft:** Eniola Sogunle.

**Writing – review & editing:** Eniola Sogunle, Gwinyai Masukume, Gill Nelson.

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
