## [Decision Letter · Decision Letter 0]

30 Aug 2019

PONE-D-19-21640

The association between caesarean section delivery and later life obesity in 21-24 year olds in an Urban South African birth cohort

PLOS ONE

Dear Dr Sogunle,

Thank you for submitting your manuscript to PLOS ONE. After careful consideration, we feel that it has merit but does not fully meet PLOS ONE’s publication criteria as it currently stands. Therefore, we invite you to submit a revised version of the manuscript that addresses the points raised during the review process.

We would appreciate receiving your revised manuscript by Oct 14 2019 11:59PM. To enhance the reproducibility of your results, we recommend that if applicable you deposit your laboratory protocols in protocols.io, where a protocol can be assigned its own identifier (DOI) such that it can be cited independently in the future. For instructions see: http://journals.plos.org/plosone/s/submission-guidelines#loc-laboratory-protocols

We look forward to receiving your revised manuscript.

Kind regards,

David Meyre

Academic Editor

PLOS ONE

Journal Requirements:

Reviewers' comments:

Reviewer's Responses to Questions

**Comments to the Author**

1. Is the manuscript technically sound, and do the data support the conclusions?

Reviewer #1: Partly

Reviewer #2: Yes

Reviewer #3: Yes

2. Has the statistical analysis been performed appropriately and rigorously? 

Reviewer #1: No

Reviewer #2: Yes

Reviewer #3: Yes

3. Have the authors made all data underlying the findings in their manuscript fully available?

Reviewer #1: No

Reviewer #2: Yes

Reviewer #3: No

4. Is the manuscript presented in an intelligible fashion and written in standard English?

Reviewer #1: Yes

Reviewer #2: Yes

Reviewer #3: Yes

5. Review Comments to the Author

Reviewer #1: Sogunle et al. present a study that examines the association between birth by Caesarean section and obesity in 889 young adult members of a South African birth cohort in Soweto, Johannesburg. The manuscript is well written but it is limited by the inability to adjust for the key confounder maternal pre-pregnancy weight. I also have some concerns with the methodology:

## Major concerns

The authors performed a post-hoc power calculation. Power calculations should always be performed *before* a study is done, not afterwards. Since the estimates from the study have an error, using them in the power calculation will result in a power estimate that also has an error and is thus meaningless. Please remove entirely.

I do not understand why the authors split the vaginal delivery group into "normal" and "assisted". From the microbiome hypothesis standpoint, there is no reason to split this group. Others have splitted the C-section group into "before 2nd stage of labor" and "at 2nd stage of labor" to capture exposure to the vaginal flora in the latter group, but I am not aware of any other paper on the CS/OB association that split the vaginal delivery group. I would strongly suggest that the authors remove that analysis and instead present the combined vaginal delivery group analysis (Table 6, top) as their main analysis.

The prevalence of obesity is 11.8% for the overall sample and 20.9% in the women. With a non-rare outcome, logistic regression is not the best choice of model because of the overestimation of the RR by the OR and the non-collapsibility of the OR (Shrier and Pang 2015). The authors should re-run the analysis with either a Poisson model with robust standard errors or a log-binomial model (Knol et al. 2012). The former is easily done in Stata using

`poisson y x1 x2 x3, robust irr`

Knol MJ, Le Cessie S, Algra A, Vandenbroucke JP, Groenwold RH. Overestimation of risk ratios by odds ratios in trials and cohort studies: alternatives to logistic regression. CMAJ.2012 May 15;184(8):895-9.

Shrier I, Pang M. Confounding, effect modification, and the odds ratio: common misinterpretations. J Clin Epidemiol. 2015 Apr;68(4):470-4.

The authors created separate categories for missing covariates, with some of these categories having n=1 obeservations. While this is a quick and easy fix, it is not appropriate anymore in this day and age where all statistical software packages have easily accessible multiple imputation commands:

https://stats.idre.ucla.edu/stata/seminars/mi_in_stata_pt1_new/

Please use MI to impute the missing values.

Why was ethnicity not included as a confounder in the models? It seems like an obvious choice, since it affects both CS rates and obesity risk.

Please remove the number needed to harm calculation, as you are not able to assume causality when one key confounder was not available (maternal pre-pregnancy weight).

There are several typos in the tables:

Table 1 - The n's for Ethnicity, Education do not add up to 889 horizontally and vertically.

Table 4 - Either the BMI median (23.3) or the IQR (20.7, 23.3) for the overall group, CS stratum are not correct. Also, the authors should check the p value for the male group - it appears to be falsely low.

The cohort originally included some 3400 people. The authors should state if and how the remaining 889 adults differ from the original cohort.

The authors claim that their study is the first to examine the association between C-section and obesity in the African context. They should discuss (in the Discussion) how and why they think the association would be different in the Southafrican context compared to high income countries or non-African middline income countries.

## Minor concerns

Table 6 could be omitted as all the relevant information is already in the text.

Line 250: The effect estimate (1.50) is outside the 95% CI (1.02, 1.20)

Line 255: The beta coefficient (0.12) is outside the 95% CI (0.01, 0.03)

Line 258: Please do not cite a commentary as evidence. There is a systematic review by the same authors that could be cited if need be:

Kuhle S, Tong OS, Woolcott CG. Association between caesarean section and childhood obesity: a systematic review and meta-analysis. Obes Rev. 2015 Apr;16(4):295-303.

Line 276: should read "seven-year" (with hyphen)

Line 289: give % instead of n ("72 women")

Reviewer #2: This study, using the data from a cohort in South Africa, investigated the association between caesarean section delivery and risk of obesity in 21-24 years. This topic is not new. However, the study design was good, and the manuscript is well written.

Page 6 line 145-148, I think it is better to remove this part in the “potential cofounders”

Page 9 line 179-187, the author says, ”compared with primary cohort,…….”, but I did not see the table or figure illustrate this result.

You can consider to merge table 3 and table 4, as they showed similar results.

Reviewer #3: The manuscript by Sogunle and colleagues describes the results of a study evaluating the relation of mode of delivery - vaginal vs. cesarean - with obesity among young adults. As pointed out by the authors, this manuscript follows a growing but consistent literature linking birth by cesarean section to increased adiposity to childhood obesity. Also as pointed out by the authors, there is less literature addressing the question of whether this risk persists later in life and there is no literature from Africa so far. This paper addresses these two important gaps. Despite the importance of providing information that can help close these knowledge gaps, the paper has some important limitations - many of them already acknowledged by the authors - that require some revisions prior considering further this manuscript. My main concerns are the following.

1. Possibility of confounding by indication. As is the case of most of the literature addressing this topic, the most important limitation of this paper is that it cannot distinguish between cesarean deliveries performed based on a clear obstetric/medical indication from cesarean deliveries performed due to maternal request, physician’s convenience, repeat cesarean or similar reasons. The introduction, however, mentions a potential instrumental variable for non-indicated cesareans. Specifically, the enormous gaps in cesarean delivery rate between hospitals in the public sector and hospitals in the private sector is unlikely to be explained by any reason different to convenience - wether it is convenience of the mother, the physician or both. Can the authors present data on mode of delivery by type of hospital in Table 1. Also, please consider type of hospital as a potential confounder and effect modifier. Also, while the authors do not have data on indication of cesarean, there is information of normal vs assisted vaginal delivery which could serve as a proxy for indication for medical intervention. In other words, the same events that lead to an instrumented vaginal delivery in South Africa, could in o different context lead to a cesarean delivery. Being able to provide different estimates for natural vs instrumented vaginal is a particular strength of this paper and I would suggest that the authors highlight further this particular strength of their paper in the discussion.

2. Possibility of selection bias. The authors mention that only a fraction of the original participants could be assessed for outcome. They also mention that the frequency of cesarean delivery does not differ between participants included and not included in the analysis. This is important as it makes it any major effect of selection bias less likely. I would suggest adding a supplemental table showing key characteristics of individuals included and not included in the analysis to further highlight this point and to mention this issue briefly in the discussion emphasizing how this may or may not result in selection bias.

3. Possibility of residual confounding. I agree with the authors that a major weakness of the study is the lack of key data on potential confounders, particularly for maternal prepregnancy BMI. While the authors do mention this as a potential issue in the discussion, quantifying the potential effect of residual confounding would be ideal. A simple and easy to interpret method to provide this quantification is the one described by VanderWeele and colleagues (Ann Intern Med 2017; JAMA 2019). I would encourge the authors to use this or other methods to quantify the effect of unmeasured/residual confounding in this paper.

4. Please justify the split of 3100g to test EM by birthweight

5. Please describe briefly the methods used to estimate the number needed to harm

6. Lines 244-259. Please also mention ref 25 in this paragraph as it appears to be the one paper not included in the meta-analysis that is not explicitly mentioned in this paragraph.

6. PLOS authors have the option to publish the peer review history of their article (what does this mean?). If published, this will include your full peer review and any attached files.

Reviewer #1: No

Reviewer #2: No

Reviewer #3: No

---

## [Author Response · Author response to Decision Letter 0]

23 Oct 2019

REVIEWERS’ COMMENTS

Please find the point-by-point responses to the reviewers’ comments below.

REVIEWER #1

Sogunle et al. present a study that examines the association between birth by Caesarean section and obesity in 889 young adult members of a South African birth cohort in Soweto, Johannesburg. The manuscript is well written but it is limited by the inability to adjust for the key confounder maternal pre-pregnancy weight. I also have some concerns with the methodology.

Major concerns

Comment 1: The authors performed a post-hoc power calculation. Power calculations should always be performed *before* a study is done, not afterwards. Since the estimates from the study have an error, using them in the power calculation will result in a power estimate that also has an error and is thus meaningless. Please remove entirely.

Response: We have removed the power calculation, as requested. 

Comment 2: I do not understand why the authors split the vaginal delivery group into "normal" and "assisted". From the microbiome hypothesis standpoint, there is no reason to split this group. Others have splitted the C-section group into "before 2nd stage of labor" and "at 2nd stage of labor" to capture exposure to the vaginal flora in the latter group, but I am not aware of any other paper on the CS/OB association that split the vaginal delivery group. I would strongly suggest that the authors remove that analysis and instead present the combined vaginal delivery group analysis (Table 6, top) as their main analysis.

Response: We agree with this criticism which echoes Reviewer 3 (comment no. 1), that we did not adequately explain the rationale behind our splitting of the vaginal delivery group into "normal" and "assisted" and the strengths thereof. We believe that our explanation justifies our reason for retaining the ‘split’ analysis. We have thus revised the texts in the methods (Page 6, Line 133-136) and discussion (Page 18, Line 369-373) sections, as follows:

“Being able to provide different estimates for normal versus assisted VD is a particular strength of this study. Infant umbilical vein cortisol and other stress molecule concentrations differ significantly between those born by normal and assisted VD, which might influence longer term offspring energy metabolism (Mears et al. 2004). Also, higher birth weight, which has been reported to be associated with CS birth, is a risk factor for instrumental VD (Aiken et al. 2014). Nevertheless, we did, in addition, analyse the combined VD data to interrogate the potential effect of differential exposure of infants to maternal vaginal and faecal microflora, implicated in the subsequent genesis of childhood obesity, between those born by CS and VD.”

Mears K, McAuliffe F, Grimes H, Morrison J. Fetal cortisol in relation to labour, intrapartum events and mode of delivery. Journal of Obstetrics and Gynaecology. 2004;24: 129–132. doi:10.1080/01443610410001645389

Aiken CE, Aiken AR, Brockelsby JC, Scott JG. Factors influencing the likelihood of instrumental delivery success. Obstet Gynecol. 2014;123: 796–803. doi:10.1097/AOG.0000000000000188

Comment 3: The prevalence of obesity is 11.8% for the overall sample and 20.9% in the women. With a non-rare outcome, logistic regression is not the best choice of model because of the overestimation of the RR by the OR and the non-collapsibility of the OR (Shrier and Pang 2015). The authors should re-run the analysis with either a Poisson model with robust standard errors or a log-binomial model (Knol et al. 2012). The former is easily done in Stata using

`poisson y x1 x2 x3, robust irr`

Knol MJ, Le Cessie S, Algra A, Vandenbroucke JP, Groenwold RH. Overestimation of risk ratios by odds ratios in trials and cohort studies: alternatives to logistic regression. CMAJ.2012 May 15;184(8):895-9.

Shrier I, Pang M. Confounding, effect modification, and the odds ratio: common misinterpretations. J Clin Epidemiol. 2015 Apr;68(4):470-4.

Response: We agree that logistic regression is not the best choice of model, and have re-analyzed the data using the Poisson regression model with robust standard errors on Page 5, line 118-121. The results are shown on page 13 line 255-262 and Table 6 on page 14. 

Comment 4: The authors created separate categories for missing covariates, with some of these categories having n=1 observations. While this is a quick and easy fix, it is not appropriate anymore in this day and age where all statistical software packages have easily accessible multiple imputation commands:

https://stats.idre.ucla.edu/stata/seminars/mi_in_stata_pt1_new/

please use MI to impute the missing values.

Response: We have imputed the missing data, as suggested. The previous OR was 1.99 (95% CI 1.00 – 3.94); the new IRR is 1.64 (95% CI 1.01 - 2.68).

Comment 5: Why was ethnicity not included as a confounder in the models? It seems like an obvious choice, since it affects both CS rates and obesity risk.

Response: Although, ethnicity was associated with both the mode of delivery and BMI, it was not included in the analysis. Because including it did not significantly change the effect estimate, in our final regression model. We have clarified this in the manuscript (see Page 6, line 126-128).

Comment 6: Please remove the number needed to harm calculation, as you are not able to assume causality when one key confounder was not available (maternal pre-pregnancy weight). 

Response: We have removed the calculation. 

Comment 7: There are several typos in the tables:

a. Table 1 - The n's for Ethnicity, Education do not add up to 889 horizontally and vertically.

Response: The numbers have been corrected in Table 3 (previously Table 1) on page 11. 

Please note: Table 1 is now table 3 because we have added Tables 1 and 2 for comparison between the primary cohort and the study participants (see response to Reviewer 2, comment 2).

b. Table 4 - Either the BMI median (23.3) or the IQR (20.7, 23.3) for the overall group, CS stratum are not correct. Also, the authors should check the p value for the male group - it appears to be falsely low.

Response: We have removed Table 4 from the report, based on Reviewer 2’s comment that Table 3 (now Table 5 on page 13) shows similar results, i.e. information on the prevalence of obesity, stratified by sex. 

Please note: Table 3 is now Table 5 because we have added Tables 1 and 2 for comparison between the primary cohort and the study participants (see response to Reviewer 2, comment 2).

Comment 8: The cohort originally included some 3400 people. The authors should state if and how the remaining 889 adults differ from the original cohort.

Response: We have now included Tables 1 and 2, comparing the characteristics of the members in the primary cohort (n = 3273) with those in our study sample (n = 889), in the results section. Most of the participants’ characteristics (including our outcome measure (BMI)) and the predictor (mode of delivery) were similar between the two groups. Other similar characteristics were: young adults’ sex, education, alcohol habit, birth weight; and mothers’ parity. Characteristics that differed, as well as the inserted tables, are briefly discussed on pages 8-10, line 190-212. 

Comment 9: The authors claim that their study is the first to examine the association between C-section and obesity in the African context. They should discuss (in the Discussion) how and why they think the association would be different in the South African context compared to high income countries or non-African midline income countries.

Response: We have added a paragraph justifying this in the discussion section on page 16, line 318-322.

“Findings from South Africa, regarding certain established potential confounders, such as socioeconomic status (SES), are different from those reported in high income countries. For example, while most high income countries have reported that high SES leads to higher body mass index, those conducted in South Africa have found that low SES was associated with higher body mass index [Cois et al. 2015; Micklesfield et al. 2013].” 

Cois A, Day C. Obesity trends and risk factors in the South African adult population. BMC Obes. 2015;2: 42–42. doi:10.1186/s40608-015-0072-2

Micklesfield LK, Lambert EV, Hume DJ, Chantler S, Pienaar PR, Dickie K, et al. Socio-cultural, environmental and behavioural determinants of obesity in black South African women. Cardiovasc J Afr. 2013;24: 369–375. doi:10.5830/CVJA-2013-069

Minor concerns

Comment 1: Table 6 could be omitted as all the relevant information is already in the text.

Response: Table 6 containing the sensitivity analysis in our previously submitted manuscript has been added as S4 Table in the supplementary information. 

Comment 2: Line 250: The effect estimate (1.50) is outside the 95% CI (1.02, 1.20)

Response: There was a typographical error in the 95% CI. It has been corrected to 1.02 - 2.20 (see line 304, page 15).

Comment 3: Line 255: The beta coefficient (0.12) is outside the 95% CI (0.01, 0.03)

Response: This has been removed. There was an error in the published manuscript which showed a significant p value and a point estimate outside the 95% CI (https://www.ncbi.nlm.nih.gov/pmc/articles/PMC6546227/). 

Comment 4: Line 258: Please do not cite a commentary as evidence. There is a systematic review by the same authors that could be cited if need be:

Kuhle S, Tong OS, Woolcott CG. Association between caesarean section and childhood obesity: a systematic review and meta-analysis. Obes Rev. 2015 Apr;16(4):295-303.

Response: The text referring to the commentary has been removed.

Comment 5: Line 276: should read "seven-year" (with hyphen)

Response: This has been corrected in line 335, page 17. 

Comment 6: Line 289: give % instead of n ("72 women")

Response: This has been corrected in line 346, page 17.

REVIEWER #2 

This study, using the data from a cohort in South Africa, investigated the association between caesarean section delivery and risk of obesity in 21-24 years. This topic is not new. However, the study design was good, and the manuscript is well written.

Comment 1: Page 6 line 145-148, I think it is better to remove this part in the “potential cofounders”

Response: The text to which the reviewer is referring is: “Although, lifestyle and behavioral characteristics, such as YAs’ diet, physical activity, smoking habits, and alcohol consumption, have been associated with obesity, it has been suggested that they are not true confounders in the analysis of the association between mode of delivery and later life obesity”. We have not removed the variables from the list, as we think it is important to mention them, as suggested by Yuan et al. 2016, and Mueller et al. 2016) (see Page 6, line 121-124). 

Mueller NT, Mao G, Bennet WL, Hourigan SK, Dominguez-Bello MG, Appel LJ, et al. Does vaginal delivery mitigate or strengthen the intergenerational association of overweight and obesity? Findings from the Boston Birth Cohort. International journal of obesity (2005). 2017;41: 497–501. doi:10.1038/ijo.2016.219

Yuan C, Gaskins AJ, Blaine AI, Zhang C, Gillman MW, Missmer SA, et al. Cesarean birth and risk of offspring obesity in childhood, adolescence and early adulthood. JAMA pediatrics. 2016 Nov 1;170(11):e162385

Comment 2: Page 9 line 179-187, the author says, ”compared with primary cohort,…….”, but I did not see the table or figure illustrate this result.

Response: We have now included the Tables 1 and 2 in the results section on Page 9-10 (see response to Reviewer 1, comment 8).

Comment 3: You can consider to merge table 3 and table 4, as they showed similar results.

Response: As Table 3 and Table 4 showed similar results regarding the prevalence of obesity, we removed Table 4 (see response to Reviewer 1, comment 7). Table 3 (now Table 5) shows the prevalence of obesity for each mode of delivery category - page 13. 

Please note: Table 3 is now table 5 because we have added Tables 1 and 2 for comparison between the primary cohort and the study participants (see response to comment 2).

REVIEWER #3

The manuscript by Sogunle and colleagues describes the results of a study evaluating the relation of mode of delivery - vaginal vs. cesarean - with obesity among young adults. As pointed out by the authors, this manuscript follows a growing but consistent literature linking birth by cesarean section to increased adiposity to childhood obesity. Also as pointed out by the authors, there is less literature addressing the question of whether this risk persists later in life and there is no literature from Africa so far. This paper addresses these two important gaps. Despite the importance of providing information that can help close these knowledge gaps, the paper has some important limitations - many of them already acknowledged by the authors - that require some revisions prior considering further this manuscript. My main concerns are the following.

Comment 1: Possibility of confounding by indication. As is the case of most of the literature addressing this topic, the most important limitation of this paper is that it cannot distinguish between cesarean deliveries performed based on a clear obstetric/medical indication from cesarean deliveries performed due to maternal request, physician’s convenience, and repeat cesarean or similar reasons. The introduction, however, mentions a potential instrumental variable for non-indicated cesareans. Specifically, the enormous gaps in cesarean delivery rate between hospitals in the public sector and hospitals in the private sector is unlikely to be explained by any reason different to convenience - whether it is convenience of the mother, the physician or both. Can the authors present data on mode of delivery by type of hospital in Table 1? Also, please consider type of hospital as a potential confounder and effect modifier. Also, while the authors do not have data on indication of cesarean, there is information of normal vs assisted vaginal delivery which could serve as a proxy for indication for medical intervention. In other words, the same events that lead to an instrumented vaginal delivery in South Africa, could in a different context lead to a cesarean delivery. Being able to provide different estimates for natural vs instrumented vaginal is a particular strength of this paper and I would suggest that the authors highlight further this particular strength of their paper in the discussion.

Response:

a. We cannot present results by type of hospital as we do not have these data. The primary cohort comprised recruited participants from public health facilities (antenatal clinics). Although we have no evidence, it is highly unlikely that any of the women went on to give birth in a private facility due to their relatively low SES.

b. We appreciate the suggestion that not having data on type of hospital could be a limitation of the study. This has now been added: “Management practices in labour and delivery units differ across hospitals and has been reported by Plough et al. (2017) to be associated with risk of primary cesarean delivery.” Line, 356-357, page 17. 

c. We have further highlighted the strength of the paper regarding the availability of data on normal and assisted vaginal delivery. See response to Reviewer 1, comment 2. 

Plough AC, Galvin G, Li Z, Lipsitz SR, Alidina S, Henrich NJ, Hirschhorn LR, Berry WR, Gawande AA, Peter D, McDonald R. Relationship between labor and delivery unit management practices and maternal outcomes. Obstetrics & Gynecology. 2017 Aug 1;130(2):358-65.

Comment 2: Possibility of selection bias. The authors mention that only a fraction of the original participants could be assessed for outcome. They also mention that the frequency of cesarean delivery does not differ between participants included and not included in the analysis. This is important as it makes it any major effect of selection bias less likely. I would suggest adding a supplemental table showing key characteristics of individuals included and not included in the analysis to further highlight this point and to mention this issue briefly in the discussion emphasizing how this may or may not result in selection bias.

Response: We have now done this. Please refer to response to Reviewer 1, comment 8.

Comment 3: Possibility of residual confounding. I agree with the authors that a major weakness of the study is the lack of key data on potential confounders, particularly for maternal prepregnancy BMI. While the authors do mention this as a potential issue in the discussion, quantifying the potential effect of residual confounding would be ideal. A simple and easy to interpret method to provide this quantification is the one described by VanderWeele and colleagues (Ann Intern Med 2017; JAMA 2019). I would encourage the authors to use this or other methods to quantify the effect of unmeasured/residual confounding in this paper.

Response: Although we agree with your concern, we did not quantify the potential effect of residual confounding by calculating the E-value because we would be making, in the context of our study, assumptions about unmeasured confounders. This has been discussed in a paper by Ioannidis et al. (2019). 

Ioannidis JPA, Tan YJ, Blum MR. Limitations and Misinterpretations of E-Values for Sensitivity Analyses of Observational Studies. Annals of Internal Medicine. 2019;170: 108–111. doi:10.7326/M18-2159

Comment 3: Please justify the split of 3100g to test EM by birth weight.

Response: We understand that it is unusual to use 3100g for classification, hence, we have removed the EM analysis for birth weight. The classification was done because the sample size for birth weight was low for some mode of delivery and BMI categories. Therefore, the median birth weight i.e. 3100g was used for classification for the EM analysis.

Comment 4: Please describe briefly the methods used to estimate the number needed to harm

Response: We have removed this analysis. Please refer to Reviewer 1, comment 6.

Comment 6: Lines 244-259. Please also mention ref 25 in this paragraph as it appears to be the one paper not included in the meta-analysis that is not explicitly mentioned in this paragraph.

Response: We have included ref 25 in the paragraph in line 311-313 on page 16.

---

## [Editor Report · Decision Letter 1]

31 Oct 2019

The association between caesarean section delivery and later life obesity in 21-24 year olds in an Urban South African birth cohort

PONE-D-19-21640R1

Dear Dr. Sogunle,

We are pleased to inform you that your manuscript has been judged scientifically suitable for publication and will be formally accepted for publication once it complies with all outstanding technical requirements.

With kind regards,

David Meyre

Academic Editor

PLOS ONE
---

## [Editor Report · Acceptance letter]

6 Nov 2019

PONE-D-19-21640R1 

The association between caesarean section delivery and later life obesity in 21-24 year olds in an Urban South African birth cohort 

Dear Dr. Sogunle:

I am pleased to inform you that your manuscript has been deemed suitable for publication in PLOS ONE. Congratulations! Your manuscript is now with our production department. 

With kind regards,

on behalf of

Dr David Meyre 

Academic Editor

PLOS ONE